# Reproductive Coercion by Intimate Partners: Prevalence and Correlates in Canadian Individuals with the Capacity to be Pregnant

**Sylvie Lévesque**[1]*, Catherine Rousseau[2], Arianne Jean-Thorn[3], Simon Lapierre[4], **Mylène Fernet**[1], Marie-Marthe Cousineau[5]

**1** Department of Sexology, Université du Québec à Montréal, Montréal, Québec, Canada, **2** Interdisciplinary School of Health Sciences, Ottawa University, Ottawa, Ontario, Canada, **3** Department of Psychology, Université du Québec à Montréal, Montréal, Québec, Canada, **4** School of Social Work, Ottawa University, Ottawa, Ontario, Canada, **5** School of Criminology, Université de Montréal, Montréal, Québec, Canada

* levesque.sylvie@uqam.ca

## Abstract

### Objectives

Despite the large body of research on violence against women, violence that specifically targets women's reproductive autonomy and control over their reproductive health, called reproductive coercion (RC), is poorly documented in Canada. The purpose of this study is to determine the prevalence of RC behaviors in an adult Canadian community sample and to explore associated factors.

### Study design

A self-report online questionnaire was administered from September 2020 to April 2021 in Quebec and Ontario, Canada. Participants were recruited via social media, sexual and reproductive health clinics, community-based anti-violence organizations, and the project's partner organizations. The questionnaire contained validated RC questionnaire items and new items drawn from previous qualitative work. The sample comprised 427 participants, mostly self-identified as women (92%), aged 18 to 55 years (M = 29.01; SD = 6.64). Descriptive analyses and binary logistic regressions were conducted using SPSS 27.

### Results

The results of this study show that 63.9% of participants reported at least one lifetime experience of RC. According to our data, contraceptive sabotage was the most common form (62.8%). Of the participants who had been pregnant, 9.8% reported control of pregnancy outcomes. Each RC category shows a different pattern of correlates. The findings also reveal that intimate partner violence (IPV) increases the likelihood of contraceptive sabotage. Moreover, the study suggests that low education level and IPV increase the risk for control of pregnancy outcomes.

**Data Availability Statement:** Since the data contain potentially sensitive information about study participants, the Université du Québec à

Montréal (UQAM) Human Research Ethics Board has only approved storage of the dataset on secure institutional servers. Any requests to access the data can be made to Université du Québec à Montréal Human Research Ethics Board: ciereh@uqam.ca ; Reference Ethics Protocol Number: 2020-3064.

**Funding:** This study was made possible by a grant awarded to the first author (S.L.) by the Social Sciences and Humanities Research Council of Canada. The funders had no role in study design, data collection and analysis, decision to publish, or preparation of the manuscript.

**Competing interests:** The authors have declared that no competing interests exist.

## Conclusion

These findings underscore the importance of RC in the lives of many Canadian individuals with the capacity to be pregnant, and they highlight certain factors that place individuals at greater risk for RC. This knowledge can inform the development of prevention efforts and clinical interventions.

## Background

Sexual and reproductive health is a fundamental human right [1]. It includes having safe and enjoyable experiences that are free from coercion, discrimination, and violence [1]. Reproductive coercion (RC) poses a significant threat to sexual and reproductive health. RC refers to behaviors that interfere with contraception and pregnancy decisions [2–4], often experienced at the hands of an intimate partners [2, 5]. Recent studies have focused predominantly on partner-perpetrated RC in heterosexual relationships [6, 7]. Accordingly, the findings show that RC is typically perpetrated by males in order to control the contraceptive and reproductive choices of adolescent girls and women [8, 9].

Following Miller's pioneering work on RC, two types of behaviors have been identified in the literature: *birth control sabotage* and *pregnancy coercion* [10]. Birth control sabotage refers to behaviors that interfere with the use of contraception [8, 11, 12], including hiding, removing or destroying contraceptive methods [8, 13]. Pregnancy coercion refers to behaviors aimed at forcing an unwanted pregnancy [3, 13, 14]. This can take the form of threats, such as having a child with another partner if she does not become pregnant, or physically hurting and coercing the female partner [8, 9, 15].

In addition to these forms of RC, recent studies have brought to light many other facets of RC, primarily qualitative studies in which participants shared their experiences in detail [7, 16]. Two moments can be identified for the occurrence of RC: before and/or during pregnancy. While the above forms focus on the period before pregnancy, some behaviors occur while the person is pregnant. Grace and colleagues have proposed items addressing abortion coercion to isolate pregnancy-promoting intent from abortion-promoting intent [17]. Nikolajski and colleagues also document behaviors aimed at preventing access to abortion in a qualitative study [17]. Those behaviors are grouped into a 3rd category, called *control of pregnancy outcomes* [7, 18].

In fact, a variety of behaviors can be understood as RC, as these behaviors are intended to direct women's reproductive trajectory, regardless of their choices or intentions [11, 18]. For example, participants have recounted manipulative behaviors such as providing false information about contraception, fertility, and abortion (e.g., lying about vasectomy or infertility, giving incorrect information about the side effects of hormonal contraceptives or the health impacts of abortion) [17, 19]. Bagwell-Gray and colleagues have added items on pregnancy avoidance which refer, for example, to taking a contraceptive method without the abusive partner's knowledge to avoid pregnancy [8]. Grace and colleagues also report that some men pressure their partners to stop using their long-term contraceptive method or switch to another method [20].

As it is an emerging field, various grey areas persist in the current conceptualization of RC. We postulate that various experiences are not considered in existing RC scales. This is the case for the imposition of the continuation or termination of the pregnancy, which refers to the control of pregnancy outcomes, or for lying and luring behaviors that aim to confuse women's contraceptive options. Including a broad range of RC experience appears critical to ensure that all forms are captured in order to reflect the experiences of various individuals.

## Prevalence of RC and associated factors

The prevalence of RC is not yet clearly established, as it varies depending on the populations studied and measures used. One study of female university students in the United States reported that 11% had experienced RC which was measured by preventing the use of birth control or condoms to prevent pregnancy or sexually transmitted infections and putting pressure to have a child [5]. Of women currently living at domestic violence shelters in the United States, 33% reported experiencing RC using the Reproductive coercion scale [8]. In that same country, the prevalence was higher for women who attended sexual and reproductive health services in low socioeconomic communities (38%) [21] and for young African American women (37.8%) [22]. A systematic literature review estimated that from 5% to 13% of young women aged 16 to 29 who attended a family planning clinic reported experiencing RC (i.e. pregnancy coercion, birth control sabotage, and abortion coercion) at least once in their lifetime [6]. To our knowledge, there are no studies measuring the prevalence of partner-perpetrated RC in Canada.

Several studies have found associations between RC and other key variables. For example, significant associations were obtained between RC and low education level [10, 23], racialization [8, 24, 25], and being single or not involved in a romantic relationship [10, 24]. In addition, women who reported having sex with both men and women were at greater risk for reporting RC by a male partner [26]. Furthermore, several studies have linked RC to other forms of intimate partner violence (IPV) (e.g., sexual violence, emotional and psychological violence, economic violence, physical violence) [8, 17, 24].

To better understand RC issues and to guide prevention efforts and intervention strategies, this study aims to document the prevalence of different types of RC in an adult Canadian community sample and to explore their correlates. Specifically, events of RC that have not yet been incorporated into the scales used in published studies are documented in order to broaden the scope. To our knowledge, this is the first study to investigate the prevalence and correlates of RC in Canadian individuals of reproductive age with the capacity to be pregnant while considering the realities of gender and sexual diversity.

## Methods

### Recruitment and data collection

Participants were recruited online via social media, using a call for participation poster. This call for participation was also shared with various organizations (e.g., sexual and reproductive health clinics, domestic violence shelters, rape crisis centers) and our partner organizations (i.e., Planned Parenthood Ottawa and Fédération du Québec pour le planning des naissances) that relayed the information to their members, in mailing lists and on their social media network. Inclusion criteria were age 18 to 55 years, has been assigned female at birth, and had sex with a cisgender man or an individual who could have made her/them pregnant. Cisgender women, trans men, non-binary individuals, and genderfluid, agender, two-spirited and questioning individuals were included in the study.

Respondents were invited to complete an online self-administered questionnaire, in French or English, from September 2020 to April 2021. All participants were first asked to read the consent form and then to consent to participate in the study by checking their choice online. Consenting to participate automatically redirected them to the online questionnaire. Considering the sensitivity of certain issues, all respondents were given a list of resources they could contact free of charge if needed. The list was provided before they responded to the questionnaire and again after completion. Respondents who completed the questionnaire could win

one of five prizes worth $50. All study procedures received ethics approval from the principal investigator's affiliated university.

In total, 493 responses were registered on the Qualtrics platform, of which 55 were withdrawn due to ineligible, duplicate, incomplete, or too short responses. Of the 438 eligible responders, 10 did not complete the initial socio-demographic questions and were excluded from the analysis. Another respondent was excluded for not entirely completing the birth control sabotage and pregnancy-related pressure scale, for a final sample of 427 participants.

## Measures

**Reproductive coercion.**   RC was measured by 18 items adapted from the Reproductive Coercion Scale [10, 19, 27, 28]. Some items were added following a review of qualitative studies of RC. The scale was slightly adjusted using reverse translation and culturally adapted to achieve the best possible equivalence, semantically and conceptually, to a French-Canadian context [29]. The questionnaire was reviewed by the project partners and sexual and reproductive health experts, and then pre-tested by four volunteers from the targeted audience. Seven items assessed contraceptive sabotage (e.g., has an intimate partner ejaculated inside your vagina when you had agreed that he would withdraw before ejaculating?), three items assessed pregnancy coercion (e.g., has an intimate partner threatened to leave you or to have a child with someone else)?), and eight items assessed control of pregnancy outcomes (e.g., has an intimate partner threatened you so that you would get an abortion?). Participants were asked to respond to each item according to whether it had occurred in their lifetime (No, 0; Yes, 1) or in the past two years (Never, 0; Rarely, 1; Occasionally, 2; Often, 3). If participants answered yes to a question about contraceptive sabotage, a sub-question targeted the perceived intent of the behavior. Participants were then asked whether, in their perception, their partner used that behavior to get them pregnant (No, 0; Yes, 1). For this analysis, we selected lifetime prevalence, with a yes/no response, without considering the perpetrator's perceived intention. Although this indicator provides us with information about the perceived nature of behavior, we believe it is not sensitive and comprehensive enough to discriminate, on its own, RC behaviors from those more associated with sexual violence. We chose to use the accepted definitions of RC and contraceptive sabotage to select items associated with this form of RC. The internal consistency of the scale as a whole and its subscales were measured by Cronbach's alpha and Mcdonald's omega. Internal consistency of the full scale was excellent ($\alpha$ = .97; $\omega$ = .97). The *contraceptive sabotage* ($\alpha$ = .68; $\omega$ = .65) and the *control of pregnancy outcomes* ($\alpha$ = .59; $\omega$ = .55) subscales showed mitigated results. This will be further discussed in the Limitations section. As for *pregnancy coercion*, the internal consistency was very low ($\alpha$ = .25; $\omega$ = .26) and, therefore findings related to this subscale will not be presented. The homogeneity of the items of this subscale was questioned and additional methodological work must be done to ensure that these items measure the same construct before presenting the results. However, the importance of measuring the phenomenon of pregnancy coercion will be addressed in the discussion. All items retained in this article are presented in Table 2.

**Intimate partner violence.**   Lifetime intimate partner violence was measured using an adapted Composite Abuse Scale (Revised)–Short Form (CASR-SF) [30]. The scale contained five items, for example, "In your life, in an intimate relationship, have you ever been insulted, despised, or humiliated by your partner?" and "In your life, in an intimate relationship, has your partner ever restricted your outings, social contacts or decisions? These actions may have been perpetrated in person or via online devices (social media, e-mail, monitoring devices, etc.)." Participants responded either No (0) or Yes (1). A dichotomous score was created to assess which participants had not experienced IPV (scored 0) and which had experienced one

or more IPV episodes (scored 1). In this study, internal consistency measured by Cronbach's alpha and McDonald's omega was satisfactory ($\alpha$ = .78; $\omega$ = .78).

**Perceived social support.** Perceived social support was measured using items drawn from the Social Provisioning Scale [31, 32]. The derived scale contained five items answered on a four-point Likert scale (Strongly disagree, 0; Disagree, 1; Agree, 2; Strongly agree, 3). Sample items are, "If something went wrong, someone would help me," and, "There are people I can count on in an emergency." A first score was calculated from the sum of the scores for all items, ranging from 5 to 20. Based on the average score (14), a dichotomous score was then created to determine who had less social support than the sample average (score of 0) and who had average or higher social support than the sample average (score of 1). In this study, internal scale consistency was excellent ($\alpha$ = .90; $\omega$ = .90).

**Sociodemographic variables.** Participants responded to several questions about their age, gender, sexual orientation, relationship status, place of birth, education, pre-pandemic occupation, and perceived economic status compared to people of same age to document their financial situation with respect to their needs. Based on previous studies [33, 34], we have expanded the relational status by proposing responses that illustrate different relational configurations. Other characteristics were also documented, such as the presence of a visible or non-visible disability (e.g., Do you consider yourself to be living with a visible or non-visible disability?), belonging to a visible minority group (e.g., In Canada, are you perceived as a member of a minority (or a racialized or ethnicized person) because of cultural or physical characteristics?), and being indigenous (First Nation, Metis, or Inuit).

**Analysis.** Binary logistic regressions were conducted using SPSS 27 (IBM Corp., 2020, Version 27.0) to identify correlates of RC, with no experience of RC as the reference category. Logistic regressions were run on lifetime experience of RC (n = 277) as opposed to RC experiences in the past two years (n = 165) to benefit from a greater statistical power. Prior to these analyses, chi-square tests were performed to compare the different socio-demographic groups according to RC experience. Only the sociodemographic variables that showed significant differences between subgroups were entered into the logistic regressions to be parsimonious [35–37], to include only relevant variables and to ensure that we have enough power to perform our analyses [38, 39], as it is usually done in the literature. However, because several studies obtained significant associations between racialized groups and RC occurrence [13, 18, 40], the variable visible minority group was included in the logistic regressions analysis. Because the variable presence of a visible or non-visible disability was marginally significant, it was also included in subsequent analyses. Based on the size of the sociodemographic groups, some variables were recoded for greater statistical power. The variable age was recoded into three groups since fewer people responded to the questionnaire between the ages of 36 and 55 (n = 68): (1) 18 to 25 years, (2) 26 to 35 years, and (3) 36 to 55 years. This recoding is guided by a concern for relatively similar group size. Sexual orientation was recoded into three categories: (1) heterosexual, (2) bisexual, and (3) homosexual, asexual, pansexual, and questioning. Similarly, the variable pre-pandemic occupation was grouped into three categories: (1) unemployed, (2) student, and (3) worker. The variable relational status was dichotomized as: (1) single, with no partner or with uncommitted partners, and (2) in a committed relationship with a main partner or more than one partner. The variable perceived economic status was also dichotomized as: (1) comfortable or sufficient, and (2) insufficient or poor.

The independent variables were entered hierarchically, this way of proceeding having been recognized for its relevance [41, 42]. The variables were grouped into systemic levels, as this conceptualization is common in the field of violence against women [43–45]. The first model included the individual variables (age, sexual orientation, perceived economic status, education, (pre-pandemic) occupation, visible minority, presence of disability). The relationship

variables, relationship status and IPV experience, were added to the second model. Perceived social support, considered as a community variable, was entered in the third and final model.

## Results

### Participants

Table 1 presents the sample's detailed descriptive characteristics. It comprised individuals assigned female at birth aged 18 to 55 years ($M$ = 29.01; $SD$ = 6.64), most of whom (92%) identified as cisgender women. Most of the sample were born in Canada (84%), 7.5% of the sample reported being a member of a visible minority and 1.9% were indigenous people (First Nations, Metis or Inuit). Most participants self-identified as heterosexual (58.5%), bisexual (19.2%) and pansexual (11.2%), while 8.7% indicated that they were in questioning regarding their sexual orientation. When asked about their pre-pandemic occupation (the survey was launched during the COVID-19 pandemic), most participants reported working (54.8%) or studying (33.7). Nearly two-thirds of the sample had completed university studies (40.3% undergraduate studies; 22% graduate studies). Most of the sample reported their economic situation as comfortable (23%) or sufficient (to meet basic needs or those of the family) (56.7%), while 11.5% reported it insufficient to meet their basic needs or indicated living in poverty (2.6%). More than three-quarters of the sample were in a committed relationship, either with a main partner (69.8%) or with more than one partner (6.1%), while 14.3% reported being single, with no partner and 9.8% indicated being single, but with one or a few partners. About one out of ten (12.2%) noted living with a visible or non-visible disability.

### Lifetime prevalence of reproductive coercion

Overall, 63.9% of participants reported at least one RC experience. Specifically, in their lifetime, 62.8% had experienced contraceptive sabotage, the most common being the partner's failure to withdraw when using the withdrawal method for contraception (41.2%), followed by being forced to have sex without a condom because the partner refused to use one (39.1%), and partner's nonconsensual removal of a condom during sex, also called stealthing (23.7%). In this sample, 9.8% of the participants who had been pregnant had experienced control of pregnancy outcomes. Table 2 presents the prevalence for each form and item.

### Correlates associated with contraceptive sabotage and control of pregnancy outcomes

**Contraceptive sabotage.** In the first model, participants who disclosed a disability (visible or non-visible) were more likely to report contraceptive sabotage than those without a disability (OR = 2.08; CI = 1.02–4.23). When the relational variables were added (Model 2), disability status was no longer significant. Moreover, participants with IPV experience (OR = 3.06; CI = 1.89–4.95) were more likely to experience contraceptive sabotage than those without IPV experience. When perceived social support was added (Model 3), IPV experience (OR = 2.93; CI = 1.80–4.76) remained significant. Perceived social support was not a significant correlate.

**Control of pregnancy outcomes.** The first model, with sociodemographic variables only, obtains one significant result for education level: compared to a university degree, participants with a high school education or college degree were more likely to report control of pregnancy outcomes (OR = 3.25; CI = 1.42–7.44). When we included relational characteristics (Model 2), education remained a significant correlate (OR = 2.80; CI = 1.19–6.58). When perceived social support was added (Model 3), education remained a significant factor: participants who perceived their financial situation as insufficient or poor were at greater risk for control of

**Table 1. Sample characteristics (N = 427).**

| Variable | N[1] | % |
|---|---|---|
| **Age** | | |
| • 18 to 25 | 152 | 35.6 |
| • 26 to 35 | 207 | 48.5 |
| • 36 to 45 | 56 | 13.1 |
| • 46 to 55 | 12 | 2.8 |
| **Gender**[*][1] | | |
| • Woman | 394 | 92.3 |
| • Man | 3 | 0.7 |
| • Non-binary | 9 | 2.1 |
| • Genderfluid | 6 | 1.4 |
| • Agender | 4 | 0.9 |
| • Two-spirited | 3 | 0.7 |
| • Questionning | 7 | 1.6 |
| **Sexual orientation** | | |
| • Heterosexual | 250 | 58.5 |
| • Homosexual | 8 | 1.9 |
| • Bisexual | 82 | 19.2 |
| • Asexual | 2 | 0.5 |
| • Pansexual | 48 | 11.2 |
| • Questionning | 37 | 8.7 |
| **Relational status** | | |
| • Single, with no partner | 61 | 14.3 |
| • Single, with one or a few partners | 42 | 9.8 |
| • In a relationship, with a main partner | 298 | 69.8 |
| • In a relationship, with more than one partner | 26 | 6.1 |
| **Place of birth** [*] | | |
| • Canada | 360 | 84.3 |
| • Outside of Canada | 43 | 10.1 |
| **Education** [*] | | |
| • High school | 16 | 3.7 |
| • College | 120 | 28.1 |
| • Undergraduate university studies | 172 | 40.3 |
| • Graduate university studies | 94 | 22 |
| **Occupation** [*] | | |
| • Student | 144 | 33.7 |
| • Worker | 234 | 54.8 |
| • Unemployed or jobless, looking for a job | 4 | 0.9 |
| • Unemployed and not looking for a job | 8 | 2.6 |
| • Other | 11 | 2.6 |
| **Perception of economic situation** | | |
| • Financially comfortable | 98 | 23 |
| • Sufficient (to meet basic needs or those of the family) | 242 | 56.7 |
| • Insufficient (to meet basic needs or those of the family) | 49 | 11.5 |
| • Living in poverty | 11 | 2.6 |
| **Visible or non-visible disability**[*] | | |
| • Yes | 52 | 12.2 |
| • No | 344 | |

(*Continued*)

**Table 1.** (Continued)

| Variable | N[1] | % |
|---|---|---|
| **Member of a visible minority** | | |
| • Yes | 32 | 7.5 |
| • No | 366 | |
| **First Nations, Metis or Inuit*** | | |
| Yes | 8 | 1.9 |
| No | 394 | |

*Note*: [1] All participants were assigned female at birth

* n may not sum up to 427 due to missing data.

pregnancy outcomes (OR = 2.70; CI = 1.14–6.42). IPV and perceived social support were marginally significant correlates in the final model (IPV: OR = 7.83; CI = 0.97–63.27; *p* = 0.054; Social support: OR = 2.44; CI = 0.98–6.06; *p* = 0.054).

Table 3 summarizes the logistic regression analysis results (Readers will find results for each model in S1–S4 Tables).

**Table 2. Detailed lifetime prevalence of reproductive coercion (N = 427).**

| Item | Lifetime RC % (n) |
|---|---|
| **Any RC** | **63.9% (273)** |
| **Contraceptive sabotage** (*n = 427*) | **62.8% (268)** |
| Has an intimate partner. . . | |
| • prevented you from using a birth control method (e.g., condom, birth control pills) or prevented you from going to the clinic to get birth control? | 20.8% (89) |
| • scared you, so you hid your birth control to prevent upsetting him/her? | 2.3% (10) |
| • removed the condom without telling you while you were having sex? | 23.7% (101) |
| • forced you to have sex without a condom because they refused to use one? | 39.1% (167) |
| • ejaculated inside your vagina when you had agreed that he would withdraw before ejaculating? | 41.2% (176) |
| • given you false information about his fertility status (e.g., led you to believe that he had a vasectomy or was infertile without it being true)? | 2.8% (12) |
| • told you not to use any birth control method because it was unhealthy, could make you infertile, or other misinformation? | 8% (34) |
| **Control of pregnancy outcomes** (*n = 190*) | **9.8% (42)** |
| Has an intimate partner. . . | |
| • threatened you so that you would get an abortion? | 6.1% (26) |
| • physically harmed you in order to induce a miscarriage (e.g., shoving you, hitting you, trying to poison you)? | 1.4% (6) |
| • given you false information about abortion in order to influence your decision, so that you would have an abortion? | 2.6% (11) |
| • prevented you from accessing an abortion, for example by preventing you from going to a clinic? | .2% (1) |
| • pressured or threatened you to give the child up for adoption after birth? | .5% (2) |
| • threatened you so that you would continue the pregnancy? | 2.1% (9) |
| • given you false information about abortion in order to influence your decision, so that you would continue the pregnancy? | 1.9% (8) |
| • brutalized you so that you would continue the pregnancy? | .7% (3) |

**Table 3. Summary of hierarchical regression analysis on two categories of reproductive coercion in relation to individual, relational and community variables (at least one-episode total score).**

| | | Contraceptive sabotage | Control of pregnancy outcomes |
|---|---|---|---|
| | | Lifetime RC | Lifetime RC |
| | | OR (95% CI) | OR (95% CI) |
| **Individual variables** | | | |
| Age | | | |
| | 18 to 25 | 0.63 (0.30–1.32) | 1.23 (0.29–5.28) |
| | 26 to 35 | 0.83 (0.42–1.62) | 1.21 (0.49–2.98) |
| | 36 to 55 (Ref) | | |
| Sexual orientation | | | |
| | Bisexual | 1.48 (0.81–2.73) | 0.57 (0.15–2.10) |
| | Homosexual, Asexual, Pansexual, or Questionning | 0.81 (0.47–1.42) | 0.59 (0.19–1.79) |
| | Heterosexual (Ref) | | |
| Economic perception | | | |
| | Insufficient or poverty | 1.74 (0.87–3.47) | 0.53 (0.17–1.67) |
| | At ease financially or Sufficient (Ref) | | |
| Education | | | |
| | High school; College | 1.27 (0.78–2.05) | **2.70 (1.14–6.42)***  |
| | University (Ref) | | |
| Occupation | | | |
| | Unemployed | 0.88 (0.34–2.30) | 2.16 (0.57–8.21) |
| | Student | 0.86 (0.51–1.44) | 1.20 (0.38–3.73) |
| | Worker (Ref) | | |
| Visible minority | | | |
| | Yes | 0.71 (0.31–1.62) | 0.98 (0.21–4.53) |
| | No (Ref) | | |
| Presence of a disability | | | |
| | Yes | 1.70 (0.82–3.52) | 1.28 (0.39–4.18) |
| | No (Ref) | | |
| **Relational variables** | | | |
| Relational status | | | |
| | In a committed relationship, with a main partner or more than one partner | 0.66 (0.34–1.30) | 0.45 (0.15–1.41) |
| | Single, with no partner or a few uncommitted partners (Ref) | | |
| Lifetime Intimate partner violence | | | |
| | Yes | **2.93 (1.80–4.76)**** | 7.83 (.97–63.27)† |
| | No (Ref) | | |
| **Community variables** | | | |
| Social support | | | |
| | No | 1.46 (0.93–2.30) | 2.44 (0.98–6.06)† |
| | Yes (Ref) | | |

Note. OR = adjusted odds ratios; CI = confidence intervals; Ref = reference category.

*** = $p < .001$

** = $p < .01$

* = $p < .05$

† = $p = 0.054$. Contraceptive sabotage: $\chi^2(13) = 45.73$, $p < .001$; Cox & Snell $R^2 = .11$ Nagelkerke $R^2 = .15$. Control of pregnancy outcome: $\chi^2(13) = 23.99$, $p < .001$; Cox & Snell $R^2 = .13$ Nagelkerke $R^2 =$.

## Discussion

The purpose of this study was to document forms and occurrences of reproductive coercion (RC) in a Canadian community sample of reproductive age. The Reproductive Coercion Scale [27] was used, supplemented with items inspired by previous qualitative work [7, 20, 33]. Lifetime prevalence for all RC items was measured, and various risk and protective correlates were identified.

Our results show that RC was a relatively common event in our participants' lives. While surprising, no data on partner-perpetrated RC are available in Canada, to our knowledge. The latest survey targeting IPV focusing on the overall Canadian population, the *Survey of Safety in Public and Private Spaces*, did not include items for RC [46]. The results show however that 44% of women who had ever been in an intimate relationship with a partner reported experiencing violence (psychological, physical or sexual) in their lifetime in the context of an intimate relationship [47].

In our study, the lifetime prevalence of RC (63.9%) is higher than in other studies, where it ranges from 8% to 30% for all RC forms combined [9, 13, 24, 28, 48]. This discrepancy could be explained by our use of a more inclusive measure to capture a broader spectrum of RC behaviors, including control of the pregnancy outcome or abortion coercion. For example, the most prevalent behavior was failing to withdraw before ejaculation, a form of contraceptive sabotage that 41.2% of participants reported. Although this form of RC has been documented in previous qualitative work [49–53], to our knowledge, it has not been directly measured in quantitative surveys. This item was added because of previous qualitative work, which revealed several such situations experienced by the participants we met. For some participants, this behavior was perceived as an attempt to get them pregnant, while for others, it was a failure to respect their consent to safe and protected sex. For the latter, they did not perceive the behavior as an attempt to make them pregnant [54].

In our study, the prevalence of all forms of contraceptive sabotage (62.8%) was a very high overall rate compared to other studies. For example, a recent study that surveyed 675 women who attended Connecticut Planned Parenthood centers reported that 16.4% of their sample had experienced some form of contraceptive sabotage [55]. Our findings on the different forms indicate that one-quarter (23.7%) of our participants experienced nonconsensual condom removal, a worrying observation. However high, this rate is somewhat lower than that found in a study of nonconsensual condom removal in 2,883 Australian women attending a sexual health clinic, which reported lifetime rates of 32% [56]. These authors describe different situational factors in the experience and interpretation of nonconsensual condom removal, for a detailed understanding of the behavior. With respect to control of pregnancy outcomes, the prevalence in our study (9.8%) was slightly lower than the 17% to 19% reported in other studies [8, 27]. This could be explained by our sample composition, which is diverse in terms of sexual orientation (where contacts with male partners can be limited) and well educated overall, both of which act as protective factors against RC [10, 23]. In addition, abortion is considered an essential health care in Canada. Pregnant individuals have access to abortion services free of charge and, starting at age 14, do not need the approval of their parent or legal guardian to receive this care. This may in part explain the lower prevalence of control related to pregnancy outcome.

Some characteristics were associated with an increased risk for reporting a particular form of RC. Participants with a high school or college degree were 2.70 times more likely to experience control of pregnancy outcomes. Because education level is generally related to financial status, low education level could mean greater economic dependence on the partner [4, 6]. In situations of economic dependence, the partner's threat to leave if the other does not become

pregnant amounts to a manipulative and controlling strategy over their reproductive health [18]. The person with capacity for pregnancy is forced to make choices that can lead to economic precarity and significant financial instability.

IPV was significantly associated with both forms of RC documented in this study. For instance, participants who reported contraceptive sabotage were almost three times more likely to report IPV (e.g., psychological, physical, or sexual). Although marginally significant (p = .054, IC = .97–63.27), our results show that being in an abusive intimate relationship also increases the likelihood of control of pregnancy outcomes. These findings are consistent with several studies that obtained strong associations between IPV and RC [5, 8, 57–59]. Given the tenuous links between these two issues and the significant risks to sexual and reproductive health that cisgender women and people with capacity for pregnancy in abusive relationships face, it would be important to include RC components to IPV measures. Currently, violent and coercive behaviors in the reproductive sphere are almost systematically absent from the available tools and questionnaires. In addition, tools are needed to help women in intimate relationships identify the presence of violence and control in terms of RC. Finally, a lack of social support may also increase the risk of experiencing coercion regarding pregnancy outcome. While marginally significant (p = .054, IC = 0.98–6.06), this trend is consistent with several previous studies on gender based violence [60, 61]. This study finds, as have previous studies, that correlates of violence occur at different ecological levels and that women's life contexts help modulate their vulnerability to experiencing RC.

As mentioned in the Method section, the results for the Pregnancy coercion items were removed because of the low internal consistency of this subscale. However, we felt it was important to take the time to discuss this form of RC here. In our sample, one out of seven participants had experienced pregnancy coercion (14.1%) (see S1 Table). Despite poor internal consistency, it is important in our opinion to discuss this category of RC because of the nature of the behaviors. These behaviors are intended to impose a pregnancy, without the persons' wish. There is increased control over their reproductive trajectory through manipulation and threats. It is highly likely that these acts of RC are part of a larger context of intimate partner violence. We believe that further empirical exploration of this form of RC is necessary so that it can be better measured and ultimately better understood.

Our findings highlight the importance of prevention efforts. Given the large number of people with capacity for pregnancy who report experiencing some form of RC, prevention activities to raise awareness of these behaviors in the general population and target the most prevalent behaviors would be essential. In addition, resources should provide additional support and information that could help these individuals reflect on their relationships, autonomy, and reproductive agency. In a second phase, more targeted prevention efforts could help individuals who are exposed to this type of violence. Based on evidence from short-term IPV intervention assessments, tailored interventions could be developed to meet the specific needs of individuals who have experienced RC [62].

Our findings also underscore the need to focus on the training needs of healthcare workers and domestic violence support workers. They should be better trained to identify RC forms and provide the appropriate support to the women as needed [63, 64]. To date, the few studies that have documented RC knowledge or perceptions in such providers indicate that they recognize that RC impedes women's reproductive autonomy and that they consider it a form of violence against women [49, 65]. Tarzia and Hegarty [18] proposes in this sense the addition of abuse to RC, which is an interesting proposal to make visible the context of control and violence.

## Limitations

This study must be considered in light of certain methodological limitations. The cross-sectional design precluded establishing causal relationships between the variables or determining whether the relationships were maintained over time. In addition, a small convenience sample was used. Considering that the participants responded on a voluntary basis, the sample is relatively homogeneous, and the results may not be representative of certain marginalized populations. It is also possible that there is an over-representation of people consulting support services since the call for participation was distributed by several community organizations. Although both trans and non-binary individuals were invited to participate, few agreed to do so. Future studies should develop more effective recruitment strategies to encourage participation by a broader range of individuals.

Furthermore, the internal consistency showed mixed results, which represents a considerable limitation of this study. Of the various publications on a reproductive coercion scale or its validation, none of these articles assessed the internal consistency of the subscales [10, 27]. Either internal consistency was not considered [27] or only internal consistency of the entire scale was reported [10, 12]. We recognize the importance of this issue and believe that the psychometric properties of this scale require further analysis to ensure that the dimensions of reproductive coercion are properly measured. In this case, the subscale referring to pregnancy coercion behaviors contains only 3 items, which may explain its very low internal consistency. However, it is essential to note that this article aims to explore all facets of this emerging phenomenon, we propose new manifestations of RC and seek to document the wide range of RC manifestations put forward by the qualitative literature.

Finally, the operationalization of RC remains in the developmental stage. Although several rigorous scales are used to measure this phenomenon [3, 12], they have not been validated in different populations or in diverse languages. Moreover, the concept of RC is still emerging, and several qualitative studies have provided a better understanding of the processes of this phenomenon and its manifestations since the initial development of these scales [7, 20, 21]. The question of intent is also relevant, as noted by other researchers [16], particularly with respect to non-consensual condom removal and withdrawal, because they do interfere with reproductive autonomy and limit women's ability to effectively protect themselves from unwanted pregnancy. Should these behaviors be conceptualized as acts of sexual violence if the perpetrator does not intend to get their partner pregnant and thus be excluded from a RC scale? If so, how can we adequately capture this intent? More research is needed to reflect on these matters and validate a new reproductive coercion scale.

Notwithstanding these limitations, our study documents RC in several groups of individuals that are minorized in the literature, namely trans and non-binary individuals, women with migratory status, racialized minorities, and those of sexual diversity. Finally, we propose broader measures of RC that account for a wider diversity of RC behaviors, as reported by many people with capacity for pregnancy in qualitative studies.

## Future research

Based on our findings, we believe it is essential to better understand the correlates of contraceptive sabotage in order to prevent this form of reproductive coercion (RC), which undermines reproductive autonomy. Whereas the data indicate that two out of three people with capacity for pregnancy report at least one such event in their lifetime, our analyses did not identify variables or contexts that could increase the risk of experiencing this form of RC. Future studies should aim to fill this significant gap in the knowledge.

Recent studies have documented RC behaviors perpetrated by individuals outside the intimate relationship, such as a family member [3, 6, 50]. This calls for the added consideration of RC that occurs outside of intimate settings. For example, in Canada, Indigenous women have reported being forced to undergo sterilization without their consent [66–68]. In other cases, women were influenced by health care personnel to decide on whether or not to pursue their pregnancy after a positive test result for Down syndrome [69]. Thus, pressure to terminate the pregnancy of a child with a disability can also be conceptualized as RC. Studies that investigate a wider range of RC behaviors and contexts would provide a more comprehensive understanding of the underlying issues. Finally, studies should be conducted in men who engage in RC behaviors to examine their motivations and perceptions. The results could inform prevention and education efforts to promote women's reproductive autonomy and sexual health.

## Supporting information

**S1 Table. Lifetime prevalence of items related to pregnancy coercion (N = 427).**
(DOCX)

**S2 Table. Bloc 1 of the hierarchical logistic regression.**
(DOCX)

**S3 Table. Bloc 2 of the hierarchical logistic regression.**
(DOCX)

**S4 Table. Bloc 3 of the hierarchical logistic regression (final model).**
(DOCX)

## Author Contributions

**Conceptualization:** Sylvie Lévesque.

**Data curation:** Sylvie Lévesque.

**Formal analysis:** Sylvie Lévesque, Catherine Rousseau, Arianne Jean-Thorn.

**Funding acquisition:** Sylvie Lévesque.

**Investigation:** Sylvie Lévesque.

**Methodology:** Sylvie Lévesque.

**Project administration:** Catherine Rousseau.

**Supervision:** Sylvie Lévesque.

**Validation:** Sylvie Lévesque.

**Writing – original draft:** Sylvie Lévesque, Catherine Rousseau, Arianne Jean-Thorn.

**Writing – review & editing:** Sylvie Lévesque, Simon Lapierre, Mylène Fernet, Marie-Marthe Cousineau.

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
