## [Decision Letter · Decision Letter 0]

6 Jun 2022

PONE-D-22-14055Reproductive Coercion by Intimate Partners: Prevalence and Correlates in Canadian Cis Women and Non-binary IndividualsPLOS ONE

Dear Dr. Lévesque,

Thank you for submitting your manuscript to PLOS ONE. After careful consideration, we feel that it has merit but does not fully meet PLOS ONE’s publication criteria as it currently stands. Therefore, we invite you to submit a revised version of the manuscript that addresses the points raised during the review process.

In your revised manuscript, please respond specifically to each comment from the reviewers. We look forward to reading the updated version.==============================

We look forward to receiving your revised manuscript.

Kind regards,

Amy Michelle DeBaets, PhD

Academic Editor

PLOS ONE

Journal Requirements:

a) Did participants provide their written or verbal informed consent to participate in this study?

Reviewers' comments:

Reviewer's Responses to Questions

**Comments to the Author**

1. Is the manuscript technically sound, and do the data support the conclusions?

Reviewer #1: Partly

Reviewer #2: Yes

2. Has the statistical analysis been performed appropriately and rigorously? 

Reviewer #1: No

Reviewer #2: Yes

3. Have the authors made all data underlying the findings in their manuscript fully available?

Reviewer #1: Yes

Reviewer #2: No

4. Is the manuscript presented in an intelligible fashion and written in standard English?

Reviewer #1: Yes

Reviewer #2: Yes

5. Review Comments to the Author

Reviewer #1: Thank you for the opportunity to review this manuscript. It is well-written and positioned to make important contributions to the RC literature. Below I have provided feedback on how to improve the authors' arguments and better position the paper in the existing RC literature. In particular, greater detail and justification of methodological and measurement decisions is warranted.

1. Title (& throughout): I’d recommend using “cisgender” rather than the abbreviated “cis.”

2. Abstract: Some of the results are phrased as though they are facts rather than findings from one analysis using a non-probability sample. These should be phrased more appropriately as findings from the given study.

3. Background, paragraph 1: I’d suggest a slight reframe of your description of RC (sentences 4-5). Are you focusing specifically on partner-perpetrated RC? If so, perhaps you should specify that in sentence 4. I’m also not sure about the statement that it “affects mainly intimate partners.” It is true that the term “reproductive coercion” is typically applied to instances of partner-perpetrated RC, but that doesn’t mean that it affects “mainly” intimate partners. That means that research has focused on partner-perpetrated RC. Perhaps you mean to say that many people have experienced RC from their partners?

4. Background, paragraph 2: I have not heard of the third “type” of RC you mention, “control of pregnancy outcomes.” Typically, RC is conceptualized across 2 dimensions: birth control/contraceptive sabotage and pregnancy coercion. The 2014 Miller text you’ve cited identifies these 2 dimensions, explaining that efforts to control the outcome of a pregnancy IS pregnancy coercion. Later, you cite the other Miller (2010) text; this article, too, conceptualizes the two dimensions without this third category. Although the Bagwell-Gray article does mention “controlling the outcome of a pregnancy,” it measured RC using just two, more general items, and it is unclear if the authors are arguing that a third type of RC should be conceptualized and measured distinctly. If the authors of this current study are arguing that their qualitative findings warrant the addition of a new and third RC dimension, this should be explicitly stated, and the distinction between the new dimension and the two existing, established dimensions should be thoroughly described.

5. Background, paragraph 3: I am not convinced that these are new/different forms of RC. These seem like experiences that would very likely fit into the established RC dimensions (contraceptive sabotage and pregnancy coercion). If you (and/or the authors cited in this paragraph) see them as distinct from these constructs, please explain how they are not captured in contraceptive sabotage and pregnancy coercion. In reviewing Table 2, it seems like many of the “control of pregnancy outcomes” items refer to abortion. Perhaps you are arguing that the established RCS does not account for these experiences? This could very well be an important and valid criticism. If this is the case, I would encourage the authors to explicitly make this argument. I would also question the title of this dimension…should it instead be conceptualized and labeled to highlight the pregnancy termination? Additionally, does it make sense to conceptualize abortion experiences as a distinct form of RC or should these items be added to the other dimensions? These arguments have implications for and beyond the current study, and I would encourage the authors to build a clear argument about the ways that RC should be conceptualized and measured.

6. Background: Are there any studies of RC in Canada? If so, it would be helpful to mention the prevalence of RC in Canada in paragraph 4. If not, perhaps explicitly state this in paragraph 4. The last sentence of the intro hints that this is one of the study’s contributions, but it is unclear whether there are other Canadian RC studies with less sexual diversity or whether this is the very first study of RC in Canada.

7. Methods, paragraph 1: Please provide a touch more detail about your sampling strategy. How were participants recruited from each of the sites listed? Who are the “partner organizations”?

8. Methods, paragraph 4: As discussed above, I am not convinced of this third category “control of pregnancy outcomes.” It seems to me that this could/should be conceptualized as 7 items assessing contraceptive sabotage and 11 items assessing pregnancy coercion. Furthermore, providing information about the validity and reliability of these measures is important and is in fact even more important since you are suggesting using new and unvalidated items to measure this construct. Please provide, at a minimum, the alpha/omega estimate for each sub-scale.

9. Methods, Analysis: I’m not sure what you mean in the second sentence. Please rephrase for clarity.

10. Methods, Analysis: Please justify your decision to exclude non-significant variables from your regression models.

11. Methods, Analysis: Please justify your decision to use hierarchical regression. This strategy has important limitations which must be considered and discussed (here in the methods and also in the discussion section). Why did you choose this strategy? What is your theoretical justification for the ways you entered the variables into the model (this is what differentiates hierarchical vs. stepwise regression).

12. Results, paragraph 1: Consider using another word in the second sentence other than “regular,” as the current phrasing is imprecise and could have negative connotations.

13. Results, paragraph 3: It is odd to refer to the correlations between RC subscales but not provide those correlations. I’d recommend adding those here to support these claims. I’m also confused by your statement that “these results do not establish the direction of the relationships.” Can you not determine the direction of the association?

14. Results, regression: Please provide the theoretical justifications for the way you have grouped these variables. Some are more obvious (sexual orientation) but would still benefit from explicit justification. Others (age, economic status) are less obvious. Why is age split in this way? How/why did you conceptualize “comfortable,” “sufficient,” “insufficient,” and “poor.”

15. Discussion, paragraph 2: This is a very high rate of lifetime RC. You have discussed it as such and provided a possible explanation. However, this unusually high prevalence combined with the poor justification for adding a third RC dimension and adding items to an established measure leaves me questioning the validity of the findings. The authors have cited a blinded qualitative study that informed their RC measures. Perhaps adding a touch of detail about the findings from that study could help to justify the added items. For example, I am compelled by the authors’ mention of failure to withdraw before ejaculation, which they support with a list of citations. Adding examples such as this to the methodological description could help get reader buy-in about the need to add item(s) to the RCS.

16. Discussion, Limitations: The statements about scale validation need to be positioned in the context of the already established and widely used RCS. Studies do exist (many of which you have cited elsewhere in your paper) that have rigorously described the development and validation of the RCS. If you are providing criticism of the RCS and arguing that new items are needed or that an entirely new scale is warranted, that is fine, but you should be explicit in your argumentation.

17. Discussion: I appreciate your comment in the last paragraph about the reproductive experiences of Indigenous women in Canada. I was surprised that the discussion section did not focus more on contextualizing RC experiences in Canada, considering that this is (I believe) the first study of RC in Canada. Please contextualize your RC prevalence findings in this setting.

Reviewer #2: Thank you for the opportunity to review this manuscript. The authors conducted a cross-sectional analysis of prevalence and correlates of reproductive coercion in a Canadian sample of women. This is a well-written manuscript that addresses some gaps in the literature, such as the experiences of this phenomenon in racialised and gender minority groups. The paper has some limitations, however, such as the convenience sample; also, the conceptual and theoretical underpinnings could be strengthened. I have offered a few suggestions below.

Background:

1) The term ‘reproductive coercion’ was coined in 2010. Recently, Tarzia and Hegarty (2021) suggested that ‘abuse’ should be added (i.e., reproductive ceorcion and abuse) to better capture the harm caused. I see the authors cite this paper and would encourage greater engagement with the terminology and concepts proposed within it.

2) A key argument by Tarzia and Hegarty is that definitions ought to recognise behavioural intent (i.e. to prevent or promote pregnancy) and impact (i.e. making a person feel controlled or scared). This has important implications for how we conceptualise and measure RCA. For example, is withdrawal about causing pregnancy through fear and control or is it about male desire for pleasure? A similar question could be posed with ‘stealthing.’ See paper by Tarzia that argues this is more aligned with sexual violence than it is RCA (Tarzia L, Srinivasan S, Marino J, Hegarty K. Exploring the gray areas between “stealthing” and reproductive coercion and abuse. Women & Health. 2020;60(10):1174-84.). Lack of conceptual clarity leads to poor measurement and I believe this analysis/evidence is limited by both. This may explain the very high prevalence of RCA – whether or not it is valid is an important question.

3) Page 5, you discuss prevalence of RCA in other analyses. It would be helpful to include some detail on the type of RCA measured in each study.

4) Page 6, “realities of sexual AND gender diversity”

Methods:

5) Page 7, more details on translation methods would be helpful. RCA is a complex concept – translating it may create even more complexities, since meanings can be lost or changed in the process. See: Wong HTH, Wang P, Sun Y, Newman CE, Vujcich D, Vaughan C, et al. Is sex lost in translation? Linguistic and conceptual issues in the translation of sexual and reproductive health surveys. Culture, Health & Sexuality. 2021:1-17.

6) Page 9, how was disability asked?

7) Page 9, Predictors suggest directionality.

Discussion

8) Page 14, you mention inclusive measures as if it were a strength. However, if the concept is poorly defined and the measures not specific enough, this may be a weakness. I refer back to comment 1 and 2.

9) Page 15, Minor language error - “Thus, insufficient…” I think you need another word here instead of ‘thus’ such as ‘in particular’ – ‘thus’ is not used correctly.

10) Page 18, more discussion is needed around the limitations of the convenience sample on validity of data. The high prevalence is likely due to both measurement error and high risk sample (recruited assault services).

11) There is quite a bit of repetition of results in the discussion. The Discussion section should include a brief summary of key findings relative to the research question and a more detailed discussion of similarities and differences between the current findings and previous research and an explanation of the current findings (why are they the same or different from previous findings, why did you observe what you observed). This is again where I think more attention could be paid to concepts/theories, including those offered by Tarzia and other theories of gender and power.

Tarzia, L., & Hegarty, K. (2021). A conceptual re-evaluation of reproductive coercion: centring intent, fear and control. Reproductive Health, 18(1), 87. doi:10.1186/s12978-021-01143-6

6. PLOS authors have the option to publish the peer review history of their article (what does this mean?). If published, this will include your full peer review and any attached files.

Reviewer #1: No

Reviewer #2: No

---

## [Author Response · Author response to Decision Letter 0]

10 Aug 2022

A table was provided in the cover letter: on one side, we have the reviewer's comment, and in the other column, our response.

Comments Response to reviewer

Done

a) Did participants provide their written or verbal informed consent to participate in this study?

p.7

We added the following sentence: “All participants were first asked to read the consent form and then to consent, or not, to participate in the study by checking their choice online. Consenting to participate automatically redirected them to the online questionnaire.” 

3. In your Data Availability statement, you have not specified where the minimal data set underlying the results described in your manuscript can be found. PLOS defines a study's minimal data set as the underlying data used to reach the conclusions drawn in the manuscript and any additional data required to replicate the reported study findings in their entirety. All PLOS journals require that the minimal data set be made fully available. For more information about our data policy, please see http://journals.plos.org/plosone/s/data-availability

We will update your Data Availability statement to reflect the information you provide in your cover letter. A data availability statement has been written and uploaded with the revised manuscript. (before the References section). Il it also presented at the end of this document.

Due to the sensitive nature of the subject matter and the possibility, albeit small, of identifying participants with the socio-demographic characteristics provided, the UQAM Ethics Board expressed concern about sharing this data. They prefer to operate with an individualized application process and supervised management of data access.

(For your information, in Canada, the process of data sharing (open data) is still in its infancy. However, it is expected that regulations will be put forward by the federal granting agencies by 2024. This will require the creation of institutional servers to store the research data, which is not yet effective. The fact that this is now being requested by scientific journals will certainly help speed up this process.)

p.11

Since this phrase was not the focus of the article, we have removed it from this article. It doesn't affect the presentation of the results or our ability to meet the stated objectives.

Thank you for the opportunity to review this manuscript. It is well-written and positioned to make important contributions to the RC literature. Below I have provided feedback on how to improve the authors' arguments and better position the paper in the existing RC literature. In particular, greater detail and justification of methodological and measurement decisions is warranted. n/a

6. Title (& throughout): 

I’d recommend using “cisgender” rather than the abbreviated “cis.” 

We have incorporated this recommendation throughout the text.

7. Abstract: 

Some of the results are phrased as though they are facts rather than findings from one analysis using a non-probability sample. These should be phrased more appropriately as findings from the given study. 

Thank you for your comment, we have added information that further contextualizes the results stated as part of this study. (p.2)

For example: “The results of this study show that 64.9% of participants reported at least one lifetime experience of RC. According to our data, contraceptive sabotage was the most common form (62.8%), followed by pregnancy coercion (14.1%).”

8. Background, paragraph 1: 

I’d suggest a slight reframe of your description of RC (sentences 4-5). Are you focusing specifically on partner-perpetrated RC? If so, perhaps you should specify that in sentence 4. I’m also not sure about the statement that it “affects mainly intimate partners.” It is true that the term “reproductive coercion” is typically applied to instances of partner-perpetrated RC, but that doesn’t mean that it affects “mainly” intimate partners. That means that research has focused on partner-perpetrated RC. Perhaps you mean to say that many people have experienced RC from their partners? 

We have modified the sentence to respond to the comment on page 4: “It is a social and health concern that many people have experienced from their intimate partners (Alexander et al., 2019; Swan et al., 2020). Recent studies have focused predominantly on partner-perpetrated RC in heterosexual relationships (Grace & Anderson, 2018; Moulton et al., 2021).”

9. Background, paragraph 2:

I have not heard of the third “type” of RC you mention, “control of pregnancy outcomes.” Typically, RC is conceptualized across 2 dimensions: birth control/contraceptive sabotage and pregnancy coercion. The 2014 Miller text you’ve cited identifies these 2 dimensions, explaining that efforts to control the outcome of a pregnancy IS pregnancy coercion. Later, you cite the other Miller (2010) text; this article, too, conceptualizes the two dimensions without this third category. Although the Bagwell-Gray article does mention “controlling the outcome of a pregnancy,” it measured RC using just two, more general items, and it is unclear if the authors are arguing that a third type of RC should be conceptualized and measured distinctly. If the authors of this current study are arguing that their qualitative findings warrant the addition of a new and third RC dimension, this should be explicitly stated, and the distinction between the new dimension and the two existing, established dimensions should be thoroughly described. 

We have reworded the paragraph to illustrate Miller and colleagues' (2010) two categories. We have added a paragraph illustrating manifestations that are not included in Miller and colleagues' conceptualization, but which correspond to RC (paragraph 2, pages 4-5).

10. Background, paragraph 3: 

I am not convinced that these are new/different forms of RC. These seem like experiences that would very likely fit into the established RC dimensions (contraceptive sabotage and pregnancy coercion). If you (and/or the authors cited in this paragraph) see them as distinct from these constructs, please explain how they are not captured in contraceptive sabotage and pregnancy coercion. In reviewing Table 2, it seems like many of the “control of pregnancy outcomes” items refer to abortion. Perhaps you are arguing that the established RCS does not account for these experiences? This could very well be an important and valid criticism. If this is the case, I would encourage the authors to explicitly make this argument. I would also question the title of this dimension…should it instead be conceptualized and labeled to highlight the pregnancy termination? Additionally, does it make sense to conceptualize abortion experiences as a distinct form of RC or should these items be added to the other dimensions? These arguments have implications for and beyond the current study, and I would encourage the authors to build a clear argument about the ways that RC should be conceptualized and measured. 

We have clarified our position in the text and have adopted the proposals of the reviewer (p.5): 

“Various grey areas persist in the current conceptualization of RC. We postulate that various experiences are not considered in existing RC scales. This is the case for the imposition of the continuation or termination of the pregnancy, or for lying and luring behaviors that aim to confuse women's contraceptive options.”

11. Background: 

Are there any studies of RC in Canada? If so, it would be helpful to mention the prevalence of RC in Canada in paragraph 4. If not, perhaps explicitly state this in paragraph 4. The last sentence of the intro hints that this is one of the study’s contributions, but it is unclear whether there are other Canadian RC studies with less sexual diversity or whether this is the very first study of RC in Canada. 

We have clarified this information by stating that there is, at this time, no RC prevalence available in Canada (p.6).

“To our knowledge, there are no studies measuring the prevalence of partner-perpetrated RC in Canada.”

12. Methods, paragraph 1: 

Please provide a touch more detail about your sampling strategy. How were participants recruited from each of the sites listed? Who are the “partner organizations”? 

Pages 6 & 7

We have modified the information presented to make the recruitment process clearer. We have specified the creation of recruitment posters to solicit participation in the study. These posters were shared on our social networks but were also shared with community organizations and partner organizations who relayed the call for participation to their members, in their mailing list and on their social networks.

“Participants were recruited online via social media, using a call for participation poster. This call for participation was also shared with various groups (e.g., sexual and reproductive health clinics, domestic violence shelters, rape crisis centers) and our partner organizations (i.e. Planned Parenthood Ottawa and Fédération du Québec pour le planning des naissances) that relayed the information to their members, in mailing lists and on their social media network.” 

13. Methods, paragraph 4: 

As discussed above, I am not convinced of this third category “control of pregnancy outcomes.” It seems to me that this could/should be conceptualized as 7 items assessing contraceptive sabotage and 11 items assessing pregnancy coercion. Furthermore, providing information about the validity and reliability of these measures is important and is in fact even more important since you are suggesting using new and unvalidated items to measure this construct. Please provide, at a minimum, the alpha/omega estimate for each sub-scale. 

Of the various publications on a reproductive coercion scale or its validation, none of these articles assessed the internal consistency of the subscales (Miller et al., 2010; Miller et al., 2014; Swan et al., 2021). Either internal consistency was not considered (Miller et al., 2010; Swan et al., 2021) or only internal consistency of the entire scale was reported (Miller et al., 2014).

In this article, the subcategories of reproductive coercion should not yet be considered subscales, as we propose new manifestations of RC. This present article seeks to document the wide range of RC manifestations put forward by the qualitative literature and aims to explore all facets of the phenomenon. 

As this is an important issue, the psychometric properties of a RC scale require further analysis to ensure that reproductive coercion is adequately measured. Therefore, the factor structure and internal consistency of this scale are the subject of a separate publication, as work has already been done on these items. It will be submitted to the journal Studies of Family Planning in August.

However, in response to the reviewer's relevant comment, the subsequent information has been added to the Method section: 

“In our study, the internal consistency of the scale was measured by Cronbach’s alpha and Mcdonald’s omega and showed excellent results (α = .97; ω = .97)”

14. Methods, Analysis: 

I’m not sure what you mean in the second sentence. Please rephrase for clarity. 

We have clarified the information in the given section on page 10.

“Logistic regressions were run on lifetime experience of RC, having been experienced by a larger proportion of the sample (n = 277).”

15. Methods, Analysis: 

Please justify your decision to exclude non-significant variables from your regression models. 

On page 10, we justified our decision by specifying that it was to include only relevant variables for the benefit of parsimony (Hosmer & Lemeshow, 2000; Stoltzfus, 2011; Tabashnick & Fidell, 2007) and to ensure that we had sufficient power to conduct our analyses (Agresti, 2007; Peduzzi et al., 1996).

16. Methods, Analysis: 

Please justify your decision to use hierarchical regression. This strategy has important limitations which must be considered and discussed (here in the methods and also in the discussion section). Why did you choose this strategy? What is your theoretical justification for the ways you entered the variables into the model (this is what differentiates hierarchical vs. stepwise regression). 

This comment came as a bit of a surprise to us, because in our disciplines (public health, sexology and psychology), hierarchical regression analyses have been favored for several years, to the detriment of stepwise regression analyses, in particular to value the contribution of previous studies in the targeted research field. Indeed, hierarchical analyses allow the validation of theoretical and empirical propositions by introducing variables according to recognized theoretical models. In this case, the ecological model, widely used in the field of violence against women, is used. 

Although both types of analysis have advantages and disadvantages, we voluntarily choose to pursue what is emphasized in our disciplines and in this area of research by favoring hierarchical regression.

However, the reviewer's comments indicate that the previous version of the article did not provide a sufficient explanation of these choices and procedures, which we have addressed in this new version. Specifically, we have added on page 12:

- A clarification of systemic levels.

- References to support our choices.

Methodological references:

Lewis, M. (2007). Stepwise versus Hierarchical Regression: Pros and Cons. Online Submission.

AARONSON, L. S. (1989). A cautionary note on the use of stepwise regression. Nursing Research, 38(5), 309-311.

Ecological model and IPV

Capaldi, D. M., Knoble, N. B., Shortt, J. W., & Kim, H. K. (2012). A systematic review of risk factors for intimate partner violence. Partner abuse, 3(2), 231-280.

Beyer, K., Wallis, A. B., & Hamberger, L. K. (2015). Neighborhood environment and intimate partner violence: A systematic review. Trauma, Violence, & Abuse, 16(1), 16-47.

Voth Schrag, R. J., Robinson, S. R., & Ravi, K. (2019). Understanding pathways within intimate partner violence: Economic abuse, economic hardship, and mental health. Journal of Aggression, Maltreatment & Trauma, 28(2), 222-242.

- The discussion was also revised in this regard. 

17. Results, paragraph 1: 

Consider using another word in the second sentence other than “regular,” as the current phrasing is imprecise and could have negative connotations. 

We have changed the term "regular" to "main" throughout the text and tables.

18. Results, paragraph 3: 

It is odd to refer to the correlations between RC subscales but not provide those correlations. I’d recommend adding those here to support these claims. I’m also confused by your statement that “these results do not establish the direction of the relationships.” Can you not determine the direction of the association? 

After revising the text, we have chosen to remove the paragraph to respond to comments made by the journal and by the reviewers. The reference to the correlations is removed without affecting our aims.

19. Results, regression: 

Please provide the theoretical justifications for the way you have grouped these variables. Some are more obvious (sexual orientation) but would still benefit from explicit justification. Others (age, economic status) are less obvious. Why is age split in this way? How/why did you conceptualize “comfortable,” “sufficient,” “insufficient,” and “poor.”

On page 12, we added the following sentence: Based on the size of the sociodemographic groups, some variables were recoded for greater statistical power.

We have added specific clarification as to the age groups created: “The variable age was recoded into three groups since fewer people responded to the questionnaire between the ages of 36 and 55 (n = 68): (1) 18 to 25 years, (2) 26 to 35 years, and (3) 36 to 55 years. This recoding is guided by a concern for relatively similar group size.” (p.12)

For economic status, we were more interested in documenting their perceived socio-economic status and the extent to which their financial situation allowed them to meet their needs. In fact, rather than having a salary range, we have a qualification of their financial comfort. Furthermore, we envisioned that many participants would still be living with their parents. In fact, their annual salary does not illustrate their socioeconomic status. We have added on page 9:

“�…� and perceived economic status in order to document their financial situation with respect to their needs.”

20. Discussion, paragraph 2: 

This is a very high rate of lifetime RC. You have discussed it as such and provided a possible explanation. However, this unusually high prevalence combined with the poor justification for adding a third RC dimension and adding items to an established measure leaves me questioning the validity of the findings. The authors have cited a blinded qualitative study that informed their RC measures. 

Perhaps adding a touch of detail about the findings from that study could help to justify the added items. For example, I am compelled by the authors’ mention of failure to withdraw before ejaculation, which they support with a list of citations. 

Adding examples such as this to the methodological description could help get reader buy-in about the need to add item(s) to the RCS. As discussed previously regarding comment # 13, the RC Scale has been widely used but was not subjected to a factor analysis to establish its validity and internal consistency, both in Miller’s original publications and in subsequent work with diverse populations. Therefore, adding new items and proposing a new conceptualization for RC could be a welcome addition in this field. 

We agree with the reviewer that it is important to rely on good measurement tools. In addition to the points already made in response to comment 13, we have also added the following to the article:

- Sample items for each of the RC categories have been added in the Methods section (p.10, sociodemographic variables).

- We have also added specifications about the process behind adding items that can also explain the differences in prevalence: “It was decided to add an item relating this behavior when our previous qualitative work revealed several situations of this type in the life experience of the participants we met. For some participants, this gesture was seen as an attempt to get them pregnant, while for others, it was a failure to respect their consent to safe sex but not to induce a pregnancy. These results also led us to question separately the perceived intention of the partner to direct the woman's reproductive trajectory. This distinction may, to some extent, explain the differences in prevalence.” (p.15, 2nd paragraph)

21. Discussion, Limitations: 

The statements about scale validation need to be positioned in the context of the already established and widely used RCS. Studies do exist (many of which you have cited elsewhere in your paper) that have rigorously described the development and validation of the RCS. If you are providing criticism of the RCS and arguing that new items are needed or that an entirely new scale is warranted, that is fine, but you should be explicit in your argumentation. 

We have contextualized on page 19 our argumentation according to the existing literature and scales: “Although several rigorous scales are used to measure this phenomenon (McCauley et al., 2017; Swan et al., 2021), they have not been validated in several different populations or in several languages. However, contraceptive realities may differ from one country or culture to another. Moreover, the concept of RC is still emerging, and several qualitative studies have provided a better understanding of the processes of this violence and its manifestations since the construction of these scales (Moulton et al., 2021). Therefore, it would be necessary to validate a new reproductive coercion scale and establish its reliability considering these qualitative findings. ”

22. Discussion: 

I appreciate your comment in the last paragraph about the reproductive experiences of Indigenous women in Canada. I was surprised that the discussion section did not focus more on contextualizing RC experiences in Canada, considering that this is (I believe) the first study of RC in Canada. Please contextualize your RC prevalence findings in this setting. 

We contextualized on page 16 the prevalence of postpartum control in the Canadian context where abortion is a recognized and accessible health care option for all: “In addition, abortion is considered an essential health care in Canada. Pregnant individuals have access to abortion services free of charge and, starting at age 14, do not need the approval of their parent or legal guardian to receive this care. This may explain the lower prevalence of control related to pregnancy outcome.”

We also added a paragraph on page 16 to present eh lack of data on partner-perpetrated RC in Canada and present some results on IPV from a Canadian populational survey.

Thank you for the opportunity to review this manuscript. The authors conducted a cross-sectional analysis of prevalence and correlates of reproductive coercion in a Canadian sample of women. This is a well-written manuscript that addresses some gaps in the literature, such as the experiences of this phenomenon in racialised and gender minority groups. The paper has some limitations, however, such as the convenience sample; also, the conceptual and theoretical underpinnings could be strengthened. I have offered a few suggestions below. 

23. Background: 

The term ‘reproductive coercion’ was coined in 2010. Recently, Tarzia and Hegarty (2021) suggested that ‘abuse’ should be added (i.e., reproductive coercion and abuse) to better capture the harm caused. I see the authors cite this paper and would encourage greater engagement with the terminology and concepts proposed within it. 

We have taken note of the suggestion. While we greatly appreciated Tarzia and Hegarty’s relevant reflections, we chose to maintain RC solely. We think that the term coercion, as conceptualized among others by Stark (2013, p.22) as “the use of force or threats to compel or dispel a particular response”, is sufficient. However, we have added a sentence in this sense in the last paragraph of discussion on page 19: “Tarzia and Hegarty (2021) proposes in this sense the addition of "abuse" to RC, which is an interesting proposal to make visible the context of control and violence.”

24. Background: 

A key argument by Tarzia and Hegarty is that definitions ought to recognise behavioural intent (i.e. to prevent or promote pregnancy) and impact (i.e. making a person feel controlled or scared). This has important implications for how we conceptualise and measure RCA. For example, is withdrawal about causing pregnancy through fear and control or is it about male desire for pleasure? A similar question could be posed with ‘stealthing.’ See paper by Tarzia that argues this is more aligned with sexual violence than it is RCA (Tarzia L, Srinivasan S, Marino J, Hegarty K. Exploring the gray areas between “stealthing” and reproductive coercion and abuse. Women & Health. 2020;60(10):1174-84.). 

Lack of conceptual clarity leads to poor measurement and I believe this analysis/evidence is limited by both. This may explain the very high prevalence of RCA – whether or not it is valid is an important question. 

This is a central thought, and we thank the reviewer for this insightful comment. We agree with Tarzia et al that nonconsensual condom removal by the male partner performed for the purpose of increasing his sexual pleasure differs from the same act performed with the intention of making his partner pregnant. (We also discussed this issue in a 2020 article.) However, we have found that it can be very difficult for the victims to state the intent associated with the perpetrated act in the many individual interviews we conducted in recent years. We therefore use the term perceived intent instead.

We agree with the reviewer that the context in which the RC behavior occurs is an important consideration: is it part of an intimate relationship marked by violence? Is the victim afraid of her partner? For this reason, we used a scale measuring IPV in this research, and examined this variable with the presence of RC.

We did assess perceived intention in this research project. For all behaviors where intent was not directly implied in the item formulation, we asked the respondent to rate the perceived intent associated with that behavior:

“If participants answered yes to a question about RC, a sub-question targeted the perceived intent of the behavior. Participants were then asked whether, in their perception, their partner used RC to get them pregnant (No, 0; Yes, 1). 

In our numerous team discussions, we considered the implications of not including in prevalence estimates acts of RC whose perceived intentions were not to become pregnant. The data collected show that women attribute very little intention to their partner to get them pregnant, regardless of the behavior taken. For example, for the item “Prevented you from using a birth control method (e.g., condom, birth control pills) or prevented you from going to the clinic to get birth control?”, 20.8 % responded yes. Of those, only 4.5 % said that the partner wanted to get them pregnant, 12.4 % did not know and 83 % said no (he did not want to get them pregnant). For the item “Told you not to use any birth control method because it was unhealthy, could make you infertile, or other misinformation?”, 8 % responded yes. Of those, 50 % said the partner wanted them to get pregnant, 41.7 % said he had no intention and 8.3% did not know. So while it is true that withdrawal could be done without the intention to get the partner pregnant (our data shows that of the 41.2 % who reported that behavior, 6 % feel that the partner wanted them to get pregnant, 85.8 % did not feel he wanted them to get pregnant and 8 5 do not know), this could also be the case for all the items of birth control sabotage. At this stage and with the data we have to rely on, we feel that perceived intent may not be sensible enough to discriminate what could be “real” RC. (These data are not shown in the article and are presented here to clarify our stance)

In light of the definitions used in our article - RC refers to behaviors that interfere with contraception and pregnancy decisions (Alexander et al., 2019; McCauley et al., 2017; S.N. Wood et al., 2020), and Birth control sabotage refers to behaviors that interfere with the use of contraception (Bagwell-Gray et al., 2021; Fleury-Steiner & Miller, 2020; Swan et al., 2021) - , we chose to include all CR behaviors, regardless of perceived intent. Consequently, for this analysis, we selected lifetime prevalence, with a yes/no response.” We are aware that this choice might allow for the documentation of actions that could be associated with sexual violence. We discussed it the Limitations section. 

Specifically, the new version of the article includes additions to address this comment:

- We have added a clarification of how perceived intent was measured in this study. (p. 9)

- We have added a rationale for choosing not to include perceived intent in our measure of CR and contraceptive sabotage. (p.9).

- We have added text to this effect in the Limitations section (p.21).

We hope that while imperfect, our decision is congruent with the actual state of knowledge on RC. We are pursuing qualitative interviews to dig further into that specific notion, in the hope of publishing later a mixed-methods paper on the perceived intent and acknowledgment of RC. Thank you for helping us shape our thoughts and reflect more deeply on this important matter.

25. Background Page 5:

you discuss prevalence of RCA in other analyses. It would be helpful to include some detail on the type of RCA measured in each study. 

We have added details of the forms of RC measured for the prevalence reported in the 4th paragraph of the background. (pp.5 & 6)

26. Background Page 6:

“realities of sexual AND gender diversity” 

On page 6, We have modified the sentence in this sense:

“To our knowledge, this is the first study to investigate the prevalence and correlates of RC in Canadian cisgender women and non-binary individuals of reproductive age while considering the realities of gender and sexual diversity.”

27. Methods Page 7:

more details on translation methods would be helpful. RCA is a complex concept – translating it may create even more complexities, since meanings can be lost or changed in the process. 

See: Wong HTH, Wang P, Sun Y, Newman CE, Vujcich D, Vaughan C, et al. Is sex lost in translation? Linguistic and conceptual issues in the translation of sexual and reproductive health surveys. Culture, Health & Sexuality. 2021:1-17. 

We thank the reviewer for the reference to this interesting article who guided us on what to include to better describe our translation and cultural adaptation process. We have added some information in the article for this purpose. 

We have also added a reference to a submitted article in which we explain the process of adapting this questionnaire by performing factorial analyses. 

The questionnaire was available in French and English because Canada is a bilingual country, while the province of Quebec is one of the only provinces with a majority of French speakers. English is the second language in Quebec, while in the rest of Canada, English is the first language.

Here is some additional information about the translation, referring to the main text: “The scale was slightly adjusted using reverse translation and culturally adapted to achieve the best possible equivalence, semantically and conceptually, to a French-Canadian context (Corbière & Fraccaroli, 2014). (1) The questionnaire was reviewed by the project partners, (2), and sexual and reproductive health experts, and then pre-tested by four volunteers from the targeted audience (3).”

(1) All three authors who have worked on this project are French-speaking and fluent in English as a second language. Two of the authors (SL and CR) have undergraduate and graduate training in sexology and are familiar with cultural references associated with sexuality, reproductive health, and violence against women. 

(2) A consultant specializing in the development and evaluation of health questionnaires.

(3) Among those consulted, there was diversity in gender expression and romantic/sexual orientation. These individuals were French speaking.

28. Methods Page 9:

how was disability asked? 

p.10

The disability question was phrased as follows: Do you consider yourself to be living with a visible or non-visible disability?

If the person answered yes, they were asked to specify which of the following: intellectual, auditory, motor, speech and language, visual, autism spectrum disorder, severe mental health disorder, other (please specify).

We have added this specification in the paragraph Sociodemographic variables in Methods.

“Participants responded to several questions about their age, gender, sexual orientation, relationship status, place of birth, education, pre-pandemic occupation, and perceived economic status to document their financial situation with respect to their needs. Other characteristics were also documented, such as the presence of a visible or non-visible disability (e.g., Do you consider yourself to be living with a visible or non-visible disability?), membership in a visible minority group (e.g., In Canada, are you perceived as a member of a minority (or a racialized or ethnicized person) because of cultural or physical characteristics?), and membership in a First Nation, Métis, or Inuit community.”

29. Methods Page 9:

Predictors suggest directionality. 

We have replaced this term with "correlates" throughout the text.

30. Discussion Page 14:

you mention inclusive measures as if it were a strength. However, if the concept is poorly defined and the measures not specific enough, this may be a weakness. I refer back to comment 1 and 2 �#23 et #24�. 

We have added various clarifications and specifications throughout the text. We believe that the article is now stronger in its conceptual and operational proposal of RC.

31. Discussion Page 15:

Minor language error - “Thus, insufficient…” I think you need another word here instead of ‘thus’ such as ‘in particular’ – ‘thus’ is not used correctly. 

The change has been made on page 16.

“The fact that insufficient or poor perceived economic situation doubled the likelihood of experiencing pregnancy coercion...”

32. Discussion Page 18: 

more discussion is needed around the limitations of the convenience sample on validity of data. The high prevalence is likely due to both measurement error and high risk sample (recruited assault services). 

We have added this sentence on page 19:

“It is also possible that there is an over-representation of people consulting support services since the call for participation was distributed by several community organizations.”

33. Discussion general: 

There is quite a bit of repetition of results in the discussion. The Discussion section should include a brief summary of key findings relative to the research question and a more detailed discussion of similarities and differences between the current findings and previous research and an explanation of the current findings (why are they the same or different from previous findings, why did you observe what you observed). This is again where I think more attention could be paid to concepts/theories, including those offered by Tarzia and other theories of gender and power.

Tarzia, L., & Hegarty, K. (2021). A conceptual re-evaluation of reproductive coercion: centring intent, fear and control. Reproductive Health, 18(1), 87. doi:10.1186/s12978-021-01143-6 

The discussion was adjusted during the review. We believe that in its current form, it no longer has superfluous repetitions. We have also mobilized the suggested reference, which adds a relevant theorizing component to the discussion of the results.

---

## [Decision Letter · Decision Letter 1]

19 Sep 2022

PONE-D-22-14055R1Reproductive Coercion by Intimate Partners: Prevalence and Correlates in Canadian Cisgender Women and Non-binary IndividualsPLOS ONE

Dear Dr. Lévesque,

Thank you for submitting your manuscript to PLOS ONE. After careful consideration, we feel that it has merit but does not fully meet PLOS ONE’s publication criteria as it currently stands. Therefore, we invite you to submit a revised version of the manuscript that addresses the points raised during the review process.

Your submission is nearly ready for publication - please respond to the final comments from the reviewer in your revised version. We look forward to your revised submission.==============================

We look forward to receiving your revised manuscript.

Kind regards,

Amy Michelle DeBaets, PhD

Academic Editor

PLOS ONE

Journal Requirements:

Reviewers' comments:

Reviewer's Responses to Questions

**Comments to the Author**

1. If the authors have adequately addressed your comments raised in a previous round of review and you feel that this manuscript is now acceptable for publication, you may indicate that here to bypass the “Comments to the Author” section, enter your conflict of interest statement in the “Confidential to Editor” section, and submit your "Accept" recommendation.

Reviewer #1: (No Response)

2. Is the manuscript technically sound, and do the data support the conclusions?

Reviewer #1: Yes

3. Has the statistical analysis been performed appropriately and rigorously? 

Reviewer #1: Yes

4. Have the authors made all data underlying the findings in their manuscript fully available?

Reviewer #1: No

5. Is the manuscript presented in an intelligible fashion and written in standard English?

Reviewer #1: Yes

6. Review Comments to the Author

Reviewer #1: Thank you for the opportunity to re-review this manuscript. I thank the authors for their responses and edits, which have addressed almost all of my concerns. I have two remaining comments, detailed below. If the authors would address these two comments, I believe this paper will be ready for publication and well-positioned to contribute to the RC literature.

1. I appreciate the edits you made relating to the existing RC literature and clarifying your argument. However, the argument is still a little unclear in a few areas. I’m not sure what you mean on page 4 when you discuss “items specifically related to the time of pregnancy.” I’m not sure if I’m misunderstanding the statement, but I think there is an opportunity to clarify this section, which reads to me as unclear and inconsistent. What exactly is the distinction between Grace et al.’s argument, that of “other authors,” and what you are proposing? Also, should the word “if” be added into the sentence beginning with “Thus”? In that same sentence, it is unclear why “control of pregnancy outcomes” is italicized. This is because I am still unsure if you are arguing that your study findings indicate the need for a third RC dimension or if you are arguing that these other authors (18) have proposed such a dimension. I am further confused when the next paragraph begins, “In addition to these three main forms of RC…” I do not believe that the existing literature establishes this third RC dimension, and the previous paragraph does not explicitly state that the current authors are proposing a third RC dimension be added. The final paragraph of this section (beginning “Various grey areas…”) does lay out this argument, but the section as a whole is confusing since the argument is not clear and seems to jump between (1) statements that the current literature proposes three dimensions and (2) statements that the authors are proposing a third RC dimension. Please edit this section to lay out a clear and consistent argument as to the existing RC literature and your contribution.

2. Thank you for discussing these issues related to RC measurement and my previous comment #13. It is true that Swan et al. (2021) only provide the internal consistency of the entire scale; this is because they propose a unidimensional scale (for the short-form RCS with an Appalachian sample), so there are no subscales to assess. Regardless, when proposing the use of subscales, it is appropriate to report internal consistency for the subscales rather than for the scale as a whole (DeVellis, 2022). Per DeVellis, Miller et al. (2014) should have reported internal consistency for the subscales rather than the full multi-dimensional scale. I appreciate the comment that there is an upcoming publication with more information about this measure and look forward to reading the measurement details provided there. However, the basic reliability of the measures used in this study is critical to the current manuscript as well, and I would suggest including the alpha/omega for the subscales here at a minimum.

DeVellis, R. F. (2022). Scale development: Theory and applications (4th edition). Sage Publishing.

7. PLOS authors have the option to publish the peer review history of their article (what does this mean?). If published, this will include your full peer review and any attached files.

Reviewer #1: No

---

## [Author Response · Author response to Decision Letter 1]

11 Oct 2022

Thank you again for your time. We greatly appreciate it, as it helps us improve the clarity and scope of our article.

Comments Response to reviewer

Review Comments to the Author

Reviewer #1: Thank you for the opportunity to re-review this manuscript. I thank the authors for their responses and edits, which have addressed almost all of my concerns. I have two remaining comments, detailed below. If the authors would address these two comments, I believe this paper will be ready for publication and well-positioned to contribute to the RC literature.

1.a) I appreciate the edits you made relating to the existing RC literature and clarifying your argument. However, the argument is still a little unclear in a few areas. I’m not sure what you mean on page 4 when you discuss “items specifically related to the time of pregnancy.” I’m not sure if I’m misunderstanding the statement, but I think there is an opportunity to clarify this section, which reads to me as unclear and inconsistent. What exactly is the distinction between Grace et al.’s argument, that of “other authors,” and what you are proposing? 

b) Also, should the word “if” be added into the sentence beginning with “Thus”? 

c) In that same sentence, it is unclear why “control of pregnancy outcomes” is italicized. This is because I am still unsure if you are arguing that your study findings indicate the need for a third RC dimension or if you are arguing that these other authors (18) have proposed such a dimension. 

d) I am further confused when the next paragraph begins, “In addition to these three main forms of RC…” I do not believe that the existing literature establishes this third RC dimension, and the previous paragraph does not explicitly state that the current authors are proposing a third RC dimension be added. 

e) The final paragraph of this section (beginning “Various grey areas…”) does lay out this argument, but the section as a whole is confusing since the argument is not clear and seems to jump between (1) statements that the current literature proposes three dimensions and (2) statements that the authors are proposing a third RC dimension. Please edit this section to lay out a clear and consistent argument as to the existing RC literature and your contribution. 

Response:

We have made changes to address the reviewer's comments.

a & b: The two paragraphs (p.4) have been rewritten. It now reads as follows:

“In addition to these forms of RC, recent studies have brought to light many other facets of RC, primarily qualitative studies in which participants shared their experiences in detail (7). For example, they have recounted manipulative behaviors such as providing false information about contraception, fertility, and abortion (e.g., lying about vasectomy or infertility, giving incorrect information about the side effects of hormonal contraceptives or the health impacts of abortion) (16, 17). Bagwell-Gray and al. have added items on pregnancy avoidance which refer, for example, to taking a contraceptive method without the abusive partner's knowledge to avoid pregnancy (8). Grace et al. also report that some men pressure their partners to stop using their long-term contraceptive method or switch to another method (18). 

Two moments can be identified for the occurrence of RC: before and/or during pregnancy. While the above forms focus on the period before pregnancy, some behaviors occur while the person is pregnant. Grace and colleagues have proposed items addressing abortion coercion to isolate pregnancy-promoting intent from abortion-promoting intent (16). Nikolajski et al. also document behaviors aimed at preventing access to abortion in a qualitative study (17). In fact, a variety of behaviors can be understood as RC as these behaviors are intended to direct women's reproductive trajectory, regardless of their choices or intentions (11, 18).”

c) We have removed the reference to "control of pregnancy outcomes" in this paragraph (see previous point).

d)The sentence has been modified. It now reads: « In addition to these forms of RC… ”

e) In addition to clarifying the points noted above, we have added the passage on control of pregnancy outcomes:

“This is the case for the imposition of the continuation or termination of the pregnancy, which constitutes a control of pregnancy outcomes, or for lying and luring behaviors that aim to confuse women's contraceptive options.” We believe this addition corrects the confusion noted by the reviewer.” (p.5)

2. Thank you for discussing these issues related to RC measurement and my previous comment #13. It is true that Swan et al. (2021) only provide the internal consistency of the entire scale; this is because they propose a unidimensional scale (for the short-form RCS with an Appalachian sample), so there are no subscales to assess. Regardless, when proposing the use of subscales, it is appropriate to report internal consistency for the subscales rather than for the scale as a whole (DeVellis, 2022). Per DeVellis, Miller et al. (2014) should have reported internal consistency for the subscales rather than the full multi-dimensional scale. I appreciate the comment that there is an upcoming publication with more information about this measure and look forward to reading the measurement details provided there. However, the basic reliability of the measures used in this study is critical to the current manuscript as well, and I would suggest including the alpha/omega for the subscales here at a minimum.

DeVellis, R. F. (2022). Scale development: Theory and applications (4th edition). Sage Publishing. 

Response:

We added the information on the internal consistency of each subscale in the Methods section (p.9) and discussed its limitations in the discussion (p.21).

---

## [Decision Letter · Decision Letter 2]

19 Oct 2022

PONE-D-22-14055R2Reproductive Coercion by Intimate Partners: Prevalence and Correlates in Canadian Cisgender Women and Non-binary IndividualsPLOS ONE

Dear Dr. Lévesque,

Thank you for submitting your manuscript to PLOS ONE. After careful consideration, we feel that it has merit but does not fully meet PLOS ONE’s publication criteria as it currently stands. Therefore, we invite you to submit a revised version of the manuscript that addresses the points raised during the review process.

 In your revised submission, please respond to the specific requests from the reviewer. We look forward to reading your updated version.

We look forward to receiving your revised manuscript.

Kind regards,

Amy Michelle DeBaets, PhD

Academic Editor

PLOS ONE

Journal Requirements:

Reviewers' comments:

Reviewer's Responses to Questions

**Comments to the Author**

1. If the authors have adequately addressed your comments raised in a previous round of review and you feel that this manuscript is now acceptable for publication, you may indicate that here to bypass the “Comments to the Author” section, enter your conflict of interest statement in the “Confidential to Editor” section, and submit your "Accept" recommendation.

Reviewer #1: (No Response)

2. Is the manuscript technically sound, and do the data support the conclusions?

Reviewer #1: Partly

3. Has the statistical analysis been performed appropriately and rigorously? 

Reviewer #1: No

4. Have the authors made all data underlying the findings in their manuscript fully available?

Reviewer #1: Yes

5. Is the manuscript presented in an intelligible fashion and written in standard English?

Reviewer #1: Yes

6. Review Comments to the Author

Reviewer #1: My only remaining concern is regarding the internal consistency of the subscales. First, you seem to have left some placeholder text on page 9: "subscales showed X results." This needs to be corrected. Second, these alphas and omegas are problematic. An argument could be made for keeping the contraceptive sabotage subscale and even the pregnancy coercion subscale. However, these measures of internal consistency may be highlighting a critical problem with your pregnancy pressure subscale. I appreciate that you have discussed this issue in your limitations section, but there needs to be more attention there. The typical cutoff is 0.7, so your values of 0.25 and 0.26 are quite worrying. The subscale either needs to be revised or you need to build a stronger argument for why it is acceptable to use it despite these incredibly low measures of internal consistency. Is there another measure of reliability that you can introduce to help justify use of your subscale?

7. PLOS authors have the option to publish the peer review history of their article (what does this mean?). If published, this will include your full peer review and any attached files.

Reviewer #1: No

---

## [Author Response · Author response to Decision Letter 2]

22 Feb 2023

Reproductive Coercion by Intimate Partners: Prevalence and Correlates in Canadian Individuals with the Capacity to be Pregnant

PLOS ONE

3rd Submission – February 2023

Dear Editor,

Again, thank you for allowing us more time for our submission. We have taken the time to modify our manuscript so that it incorporates the comments made during its last revision.

Two comments were made: 

Reviewer #1: My only remaining concern is regarding the internal consistency of the subscales. First, you seem to have left some placeholder text on page 9: "subscales showed X results." This needs to be corrected.

- Thank you for noticing this mistake, we corrected it in the manuscript.

Second, these alphas and omegas are problematic. An argument could be made for keeping the contraceptive sabotage subscale and even the pregnancy coercion subscale. However, these measures of internal consistency may be highlighting a critical problem with your pregnancy pressure subscale. I appreciate that you have discussed this issue in your limitations section, but there needs to be more attention there. The typical cutoff is 0.7, so your values of 0.25 and 0.26 are quite worrying. The subscale either needs to be revised or you need to build a stronger argument for why it is acceptable to use it despite these incredibly low measures of internal consistency. Is there another measure of reliability that you can introduce to help justify use of your subscale?

- Following emails to the editor, we decided to present this subscale in the Method section and point out that the reliability is very low and therefore, not present the findings in the Results section. However, we still wanted to address these behaviors in the Discussion section (as seen on page 18). We believe that this will allow other researchers to continue thinking about this topic, and to further document the different forms of reproductive coercion, while allowing us and PLOS One to publish an article that meets current scientific standards. 

- We have chosen to present the items related to pregnancy coercion as Supplementary materials. From our perspective, we believe that the complete elimination of these items does not serve the goals of documenting and understanding this emerging phenomenon. Indeed, current work in qualitative research shows that some behaviors are not documented by existing reproductive coercion scales comprising subscales of contraceptive sabotage and control of pregnancy outcomes. 

We would also like to bring another point to your attention. In the original submission, we presented the three broad categories of reproductive coercion. One of the reviewers asked us to propose a double categorization, arguing that all three categories were not recognized in the scientific literature. We have modified our proposal, in response to the journal's requests. However, we are not comfortable with this way of presenting the accumulated knowledge in this area, as we believe it is outdated. Recent major publications on this topic, including a qualitative synthesis of evidence published in December 2021 in your journal https://doi.org/10.1371/journal.pone.0261551, clearly present the three main forms that we were documenting. We chose to return to the essence of our original proposal, but in a clearer presentation, as the reviewer was right to point out some incoherencies in our original submission. This paragraph is presented on page 4. 

We hope this revised version will meet your expectations.

Sylvie Lévesque (elle)

Professeure

Département de sexologie

Université du Québec à Montréal

WR-430

 514-987-3000 poste 5816

@ levesque.sylvie@uqam.ca

Université du Québec à Montréal 

Case postale 8888, succursale Centre-ville

Montréal (Québec) H3C 3P8

---

## [Editor Report · Decision Letter 3]

6 Mar 2023

Reproductive Coercion by Intimate Partners: Prevalence and Correlates in Canadian Individuals with the Capacity to be Pregnant

PONE-D-22-14055R3

Dear Dr. Lévesque,

We’re pleased to inform you that your manuscript has been judged scientifically suitable for publication and will be formally accepted for publication once it meets all outstanding technical requirements.

Kind regards,

Amy Michelle DeBaets, PhD

Academic Editor

PLOS ONE
---

## [Editor Report · Acceptance letter]

9 Mar 2023

PONE-D-22-14055R3 

Reproductive Coercion by Intimate Partners: Prevalence and Correlates in Canadian Individuals with the Capacity to be Pregnant 

Dear Dr. Lévesque:

I'm pleased to inform you that your manuscript has been deemed suitable for publication in PLOS ONE. Congratulations! Your manuscript is now with our production department. 

Kind regards, 

on behalf of

Dr. Amy Michelle DeBaets 

Academic Editor

PLOS ONE